

# Flux-closure domains in high aspect ratio electroless-deposited CoNiB nanotubes

Michal Staňo[1⋆], Sandra Schaefer[2], Alexis Wartelle[1], Maxime Rioult[3], Rachid Belkhou[3], Alessandro Sala[4], Tevfik O. Menteş[4], Andrea Locatelli[4], Laurent Cagnon[1], Beatrix Trapp[1], Sebastian Bochmann[5], Sylvain Y. Martin[6,1], Eric Gautier[6], Jean-Christophe Toussaint[1], Wolfang Ensinger[2] and Olivier Fruchart[6†]

**1** Univ. Grenoble Alpes, CNRS, Institut NEEL, F-38000 Grenoble, France
**2** Technische Universität Darmstadt, D-64287 Darmstadt, Germany
**3** Synchrotron SOLEIL, Saint-Aubin, F-91192 Gif-sur-Yvette, France
**4** Elettra - Sincrotrone Trieste S.C.p.A., I-34149 Basovizza, Trieste, Italy
**5** Friedrich-Alexander University Erlangen-Nürnberg, D-91058 Erlangen, Germany
**6** Univ. Grenoble Alpes, CNRS, CEA, Grenoble INP, INAC – Spintec, F-38000 Grenoble, France

⋆ michal.stano@atlas.cz, † olivier.fruchart@cea.fr

## Abstract

We report the imaging of magnetic domains in ferromagnetic CoNiB nanotubes with very long aspect ratio, fabricated by electroless plating. While axial magnetization is expected for long tubes made of soft magnetic materials, we evidence series of azimuthal domains. We tentatively explain these by the interplay of anisotropic strain and/or grain size, with magneto-elasticity and/or anisotropic interfacial magnetic anisotropy. This material could be interesting for dense data storage, as well as curvature-induced magnetic phenomena such as the non-reciprocity of spin-wave propagation.



# 1   Introduction

Magnetic nanotubes, less reported than the solid nanowire geometry, have been considered mainly in the context of biomedicine [1] and catalysis [2]. In nanomagnetism and spintronics, mainly planar strips prepared by lithography and more recently solid cylindrical nanowires have been investigated as one-dimensional conduits for the motion of magnetic domain walls. Besides the fundamental interest, these are mentioned as possible candidates for novel data storage devices, mainly focusing on the concept so-called race-track memory [3,4] based on shifting magnetic domain walls with spin-polarized current. Strips are easier to fabricate with a large versatility, while wires and tubes open opportunities to new physics of three-dimensional textures and curvature-induced effects.

In case of nanotubes, theory and simulations predict similar physics of magnetic domains and domain walls compared to cylindrical nanowires, most interestingly in dynamics, with high domain wall velocity and interaction with spin waves [5]. However, the potential of tubes for new physics and devices is higher than that of nanowires. Indeed, their magnetic properties can be tuned by changing the tube wall thickness [6] and more complex architectures can be prepared based on core-shell structures [7], analogous to multilayers in 2D spintronics. Further, the curvature is associated with breaking of an inversion symmetry, as the inner and the outer surfaces are not equivalent, providing an analogy with multilayered flat films/strips in which breaking of the inversion symmetry is associated with promotion of chiral magnetic textures, fast propagation of magnetic domain walls [8], and non-reciprocity of spin wave propagation [9]. Indeed, similar phenomena have been predicted in magnetic nanotubes: curvature induces magnetochirality [10], anisotropy and a so-called effective Dzyaloshinskii-Moriya in-

teraction [11]. The most exciting situation is that of domains with azimuthal magnetization, with theoretical predictions of the non-reciprocity of spin wave propagation [12, 13].

So far, none of the above could be addressed experimentally, due to lack of a suitable material. In particular, for magnetically-soft nanotubes (i.e. considering only exchange and magnetostatic energy) calculations show that the azimuthal state is the ground state only for short tubes with a large diameter (small aspect ratio: length/diameter) and large tube wall thickness [14–16], all to be compared with the dipolar exchange length. High aspect ratio nanotubes should display axial (longitudinal) magnetization due to dominance of shape anisotropy, with azimuthal magnetization possibly found only at tube ends as a so-called end-curling state [15–17]. Experimental investigations of single nanotubes are scarce, particularly with magnetic imaging to determine in detail their magnetization state. Recently, the above micromagnetic picture could be confirmed experimentally by Wyss and coworkers [18]. Thus, azimuthal domains have been so far obtained in tubes with micrometric diameters [19] or short lengths (length of 1-2 microns for diameter around 300 nm) [18]. However, both for studies of spin wave physics and for applications (such as magnonic waveguides, data storage elements – racetrack memory) one would need longer tubes with higher aspect ratios. In this manuscript we unlock this limitation, reporting the synthesis and magnetic imaging of CoNiB nanotubes by polarized X-rays, and showing that these tubes can host azimuthal domains for long (high aspect-ratio) tubes.

## 2 Synthesis and structural analysis

Arrays of tubes had been prepared previously by several groups using techniques such as electroplating [20], atomic layer deposition (ALD) [6, 21], physical deposition with a tilted evaporation beam on vertical pillars [18], or rolling thin sheets (micrometric diameters) [22]. These studies revealed interesting features, however also limitations for the above techniques: wire-versus-tube growth instabilities for electroplating [23], granular and magnetically-imperfect material for ALD [7], not a continuous tube for rolled sheets, and physical deposition cannot be up-scaled for the fabrication of a dense vertical array of tubes.

Here we fabricate CoNiB nanotubes by conformal electroless plating inside porous ion track-etched polycarbonate membranes (pore diameter around 300 nm and length 30 microns) according to ref. [24] (details can be found also in Appendix A). This electrochemical technique provides a robust control over the tube thickness (proportional to the plating time) [25] as the material grows radially starting from tiny Pd catalysts on the pore walls (see Appendix Fig. 5). The deposition is based on reduction of metallic ions from a solution by means of an additional chemical, so-called reducing agent (dimethylamine borane in our case) that provides the electrons for the reduction. We took care to control the reaction kinetics, limiting the deposition rate (about 1.5 nm/min) to allow time for diffusion of chemical species inside the pores and thus deliver tubes with uniform wall (shell) thickness along their length, despite their high-aspect ratio. While similar plating had been already used to prepare arrays of magnetic tubes [2, 25, 26], here we synthesize and image a different material, namely nanocrystalline ($Co_{80}Ni_{20}$)B, that we will prove to reveal novel magnetic flux-closure (azimuthal) domains.

For the investigation of isolated tubes, the polycarbonate template is dissolved in dichloromethane and tubes are transferred onto a suitable substrate (technique dependent, see Appendix A.5) and in some cases aligned with an external magnetic field (Appendix A.6). The prepared CoNiB tubes have diameter 300-400 nm, length up to 30 μm (both given by the template), and tube wall thickness approximately 30 nm (given by the deposition time). The tubes are nanocrystalline (Fig. 1a,b) with a complex and hierarchical microstructure (Fig. 1c,d): 1-2 nm thick boundaries separate 10 nm grains, themselves displaying an internal structure at

the scale of 2 nm. The grain boundaries appear bright in conventional transmission electron microscopy (c) and dark in the dark-field mode (d). This highlights lighter elements at grain boundaries, such as boron and oxygen. Similar microstructure, with metallic *macrograins* embedded in a boron-rich matrix, has been already reported in case of NiB nanoparticles [27] prepared also by electroless plating with boron-containing reducing agent. Chemical composition of our tubes is further discussed in Appendix C.1.

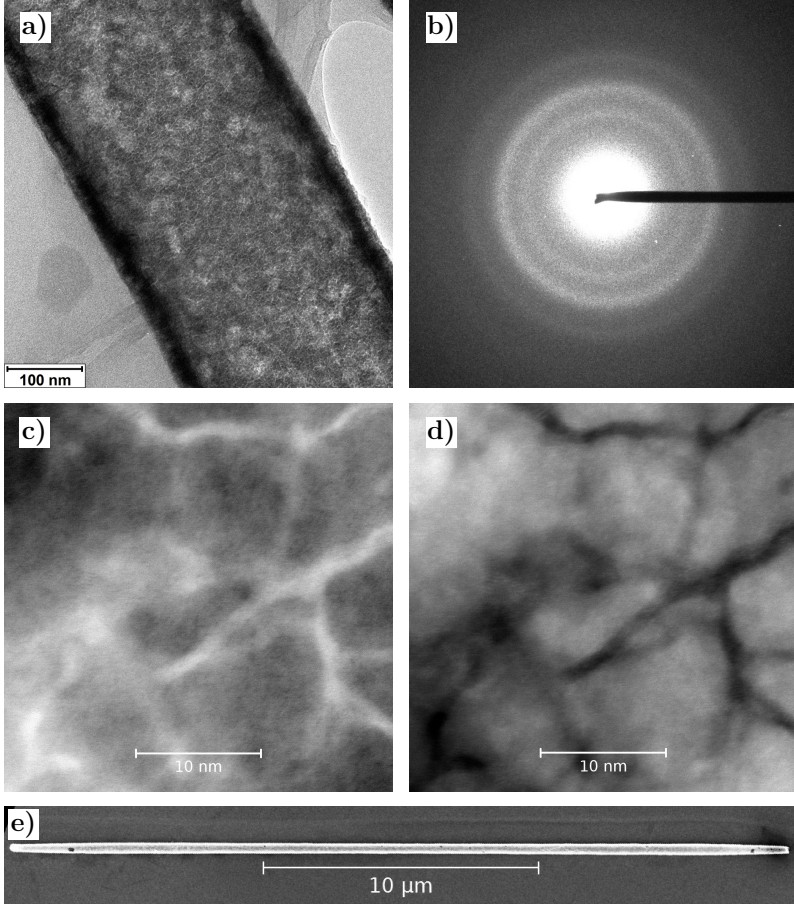

Fig. 1: **Structure of electroless-deposited CoNiB nanotubes**. a) Transmission electron microscopy image of a nanocrystalline CoNiB tube and b) corresponding selected area (240 nm in diameter) electron diffraction pattern showing diffusive rings originating from nanograins with all possible crystallographic orientations. c) Closer look on the grains with scanning transmission electron microscopy in bright and d) dark field (Z contrast, heavier elements appear brighter). e) Scanning electron microscopy image of a whole tube.

## 3 Evidence for azimuthal magnetic domains

We use X-ray Magnetic Circular Dichroism - PhotoEmission Electron Microscopy (XMCD-PEEM) to reveal the magnetic domains in the tubes. This photon-in, electron-out technique maps the component of magnetization parallel to the X-ray beam propagation direction, with a spatial resolution of 30-40 nm. We use the so-called shadow geometry on single (isolated) tubes dispersed on a doped Si substrate, as pioneered by Kimling et al. [7] and further developed in our group [28]. This method provides information about magnetization both on the tube sur-

face and in the tube volume. The latter is inferred from the photoelectron signal in the tube shadow, which reflects the magnetization-dependent dichroic X-ray transmission through the tube. Further information on X-PEEM can be found in Appendix B.1.

Fig. 2a displays an XMCD-PEEM image of two orthogonal tubes on a Si substrate. The magnetic contrast is insignificant for the tube aligned parallel to the X-ray beam direction, while it is much stronger when the beam is transverse to the tube axis. Thus, magnetization is not axial as expected from theory for long soft magnetic tubes [16], but it is perpendicular to the tube axis. Examination of the shadow reveals an inversion of contrast for X-rays having gone through the top and bottom parts of the tube (Fig. 2b), whereas uniform transverse magnetization would give rise to a monopolar contrast [28]. This proves that magnetization is not uniformly transverse in the tubes but azimuthal, curling around the tube axis. We investigated in total tens of tubes with various beam directions, all supporting this analysis. Note that the tube is multidomain: the sense (sign) of the circulation of the flux closure alternates along the tube axis.

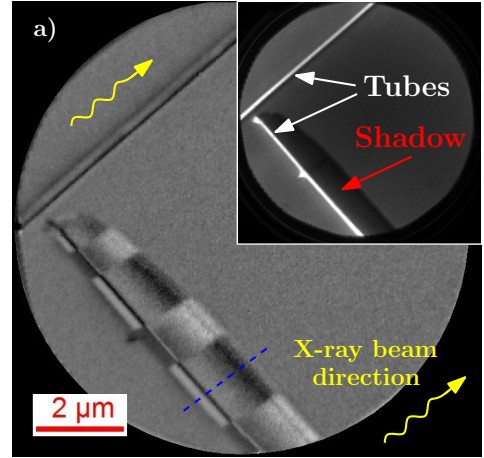

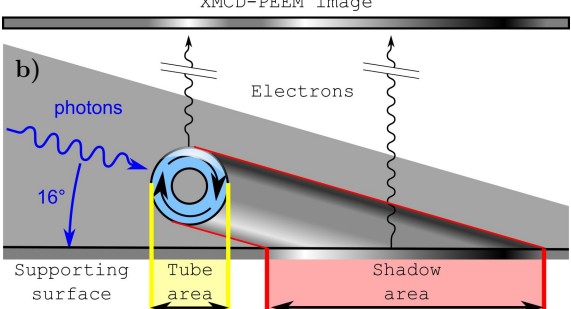

Fig. 2: **Magnetic azimuthal flux-closure domains.** a) XMCD-PEEM (Co-L$_3$ edge) image of two orthogonal tubes. The tube along the beam (top) gives rise to almost zero contrast, whereas strong contrast is observed for the tube perpendicular to the beam, revealing domains with azimuthal magnetization. The grey line in the shadow close to the rim comes from oxidation of the inner tube surface (nonmagnetic). The inset shows a non-magnetic photoemission electron microscopy image of the tubes. b) Scheme with the azimuthal magnetization and XMCD-PEEM contrast corresponding to a line profile of an azimuthal domain marked by a blue, dashed line in a). Note that in the experiment the tubes lie on the substrate and only part of the shadow can be observed. Sometimes contrast inversion can be seen also in the tube area as detailed in [28].

We find azimuthal domains only, following either AC-field demagnetization along the

transverse direction, or saturation along the tube axis. Therefore, azimuthal curling seems to be the ground state for this material and geometry. Recently, Wyss and coworkers [18] observed with XMCD-PEEM CoFeB and NiFe nanotubes (around 300 nm in diameter, 30 nm tube wall thickness, formed by sputtering on semiconducting nanowires with a hexagonal cross-section). They found also azimuthal domains (*global vortices*), however only for tubes shorter than 1-2 μm (our tubes have a length 20-30 μm); longer tubes displayed axial magnetization with the curling only at the tube ends as expected from theory [16]. Similarly, our tubes should be axially-magnetized as already discussed above. Therefore, we argue that an additional contribution, magnetic anisotropy, has to be present to promote alignment of magnetization in the azimuthal direction.

# 4 Discussion on the magnetic anisotropy

Here we provide arguments for describing and extracting the strength of the microscopic magnetic anisotropy favouring azimuthal magnetization in tubes. The first question to address is the functional form relevant to describe the volume density of magnetic anisotropy, as none of the three local directions are equivalent (radial $\hat{r}$, azimuthal $\hat{\phi}$ and axial $\hat{z}$). Given the large aspect ratio of our tubes, we assume that the local shape anisotropy (i.e. magnetostatic energy) is the dominant energy term. This is also justified later by showing that the anisotropy field determined from hysteresis loops is small compared to the spontaneous induction (tens of mT versus about 1 T, respectively). So, in the following we suppose zero radial magnetization ($m_r = 0$) in magnetic domains ($m = \frac{M}{M_s}$). Thus, describing the anisotropy with terms $-K_\phi m_\phi^2$ or $K_\phi m_z^2$ should be equivalent, because $m_\phi^2 + m_z^2 \approx 1$. Note that a positive anisotropy coefficient $K_\phi > 0$ favours azimuthal magnetization.

On the basis of the moderate wall thickness (30 nm compared to 300 nm diameter for our CoNiB tubes; in general valid for thin-walled tubes), we assume that radius-dependent variations are averaged out and taken into account in an effective uniform value of $K_\phi$. The first contribution to $K_\phi$ is magnetic anisotropy related to the (crystal) lattice $K_{\mathrm{mc}}$: magnetocrystalline, magnetoelastic or interface anisotropy (to be discussed later in the text). The second contribution is related to the exchange energy, whose volume density reads, for $m_r = 0$ [16, 29]: $E_{\mathrm{ex}} = (A/R^2)m_\phi^2$ with $A$ being the exchange stiffness and $R$ the tube radius. This term acts as a curvature-induced anisotropy : a spatial variation of magnetization exists for a uniform $m_\phi$ due to the non-uniformity of $\hat{\phi}$. Uniform $m_z$ is associated with no spatial variation, so it does not contribute. The exchange contribution favors axial magnetization in nanotubes with small diameters [16]. So finally, the total anisotropy coefficient is $K_\phi = K_{\mathrm{mc}} - A/R^2$. This can be converted into an anisotropy field $H_K = 2K_\phi/(\mu_0 M_s)$. Measuring the latter experimentally allows one to estimate the microscopic anisotropy energy coefficient: $K_{\mathrm{mc}} = A/R^2 + \mu_0 M_s H_K/2$.

We estimated $H_K$ based on a series of Scanning Transmission X-ray Microscopy (STXM, see Appendix B.2) images acquired under different external magnetic fields applied along the tube axis (Fig. 3). Upon increasing the field, the domain contrast decreases, which shows that magnetization gradually rotates towards the axial direction. Therefore, this corresponds to a hard axis loop, slanted and with zero remanence (like in Appendix Fig. 9). In such case the anisotropy and saturation fields are closely related. However, it is difficult to extract quantitatively the direction of magnetization in this series, because of the exponential decay of photon intensity inside matter, uncertainties in the dichroic coefficient, and the existence of a background intensity in the image. We can only provide an estimate of $H_K$ from the field for which all contrast vanishes in the corresponding images. We find $\mu_0 H_K \approx 25$ mT. Note that at remanence the tubes return to a flux-closure domain pattern (with close-to-zero remanence) and that the series with the opposite direction of applied field are very similar.

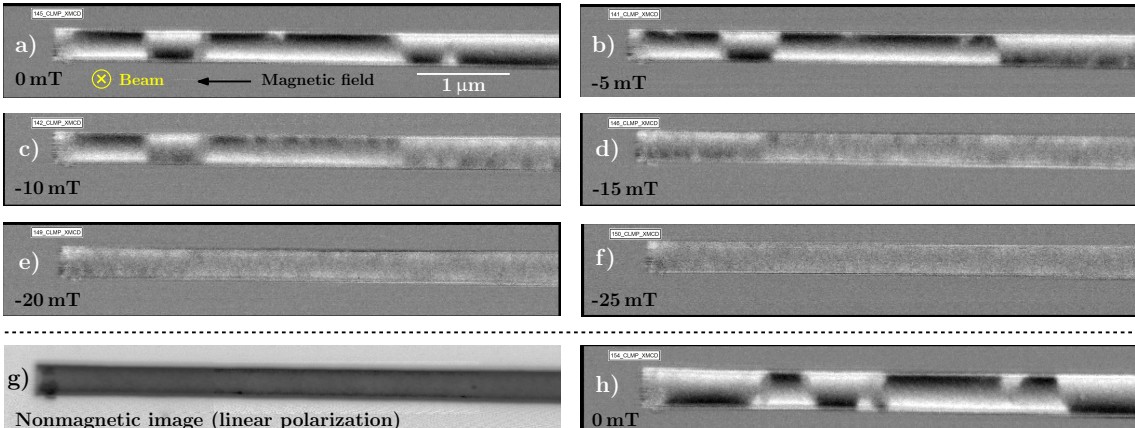

Fig. 3: **STXM under external magnetic field - anisotropy strength determination**. a)-f) XMCD magnetic images (Co-$L_3$ edge, same contrast range 15%) under axial magnetic field. With increasing the field magnitude the STXM contrast vanishes, showing that magnetization rotates towards the axial direction. Around 25 mT is needed for the saturation of tubes along the axial direction. Field of view a)-g) $6.0 \times 1.0\,\mu m^2$ and h) $4.8 \times 0.8\,\mu m^2$. g) Non-magnetic STXM image (linear polarization of X-rays) highlighting the tubular structure. h) XMCD image after removing magnetic field (after sequence a-f). Even at zero field, the transition between neighbouring domains is not as sharp as in XMCD-PEEM images; this we attribute to sample ageing (STXM done 1 year after X-PEEM).

As regards the conversion of $H_K$ to the anisotropy coefficient, we do not have a direct measurement of the exchange stiffness of our material, however, for example $Co_{80}B_{20}$ has $A \approx 10\,pJ/m$ [30]. This value may be different in our case, but the order of magnitude should be correct. Besides, the tube diameter is still large, so that the exchange penalty correction to the anisotropy is rather small, less than few mT of equivalent field, which should be well within the error/spread of the experimentally determined anisotropy field. Further, similar to curvature-induced anisotropy, curvature-induced effective Dzyaloshinskii-Moriya interaction is expected to be negligible as it also scales with the curvature, being pronounced only for tube diameters below 100 nm. Therefore, we arrive at an estimate of the anisotropy constant $K_{mc} \approx 10\,kJ/m^3$. Note, however, that this value may be affected by a sample ageing. The one-year-old sample used in STXM (rather fresh sample was used for XMCD-PEEM), shows less sharp azimuthal domains and a weak axial component of the magnetization.

Hysteresis loops on arrays of tubes (Appendix Fig. 8) obtained by global magnetometry show a different behaviour, which could be attributed to the contribution of remaining parts of the film on top/bottom of the template housing the tube array. Difference between the behaviour of tube array and isolated tubes probed by synchrotron microscopies can also mean that the anisotropy may not originate only from the growth itself, but there could be also an effect of the template dissolution and laying the tubes on the substrate. Isolated tubes on a Si substrate were probed also by magneto-optics with a focused laser (spot size $1\,\mu m$) – some of the loops (Appendix Fig. 9) were slanted with zero remanence and thus consistent with synchrotron imaging (Fig. 3), however, others were almost squared. We assume that this comes from a local heating by the focused laser, where the difference in the loops is given by the thermal contact with the substrate and thus varying evacuation of the heat.

Regarding the microscopic reason for the anisotropy, we can rule out a magneto-crystalline contribution, because of the nanocrystalline nature of the material without any preferred crystallographic orientation (see diffraction rings in Fig. 1b). Possible scenarios include inter-grain

surface magnetic anisotropy and magneto-elastic coupling (inverse magnetostriction) associated with a curvature-related anisotropy effects lifting the degeneracy between the azimuthal and axial directions. Both phenomena could yield anisotropy values compatible with the experiment (Appendix D).

For comparison we also considered nanocrystalline tubes with very similar geometry, however from material $(Ni_{80}Fe_{20})B$. These proved to be axially magnetized (Appendix E, Appendix Fig. 10). A difference between the two materials is the strength of the magnetostriction, which is sizeable and negative for $(Co_{80}Ni_{20})B$ [31] (also Appendix Tab. 1), and nearly vanishing and positive for $(Ni_{80}Fe_{20})B$ [31]. However, no information is available on surface anisotropy, so that we cannot unambiguously point at the reason for azimuthal anisotropy.

## 5 Tuning the material through annealing

We annealed the tubes at various temperatures and examined their magnetization state after cooling to room temperature. As can be seen in Fig. 4, the XMCD-PEEM contrast associated with the azimuthal domains becomes weaker and finally disappears with increasing annealing temperature.

We attribute the loss of contrast to a gradual rotation of magnetization towards the axial direction. The final weak uniform contrast is determined by the small projection of the axial magnetization to the beam direction (the beam is slightly away from the perpendicular to the

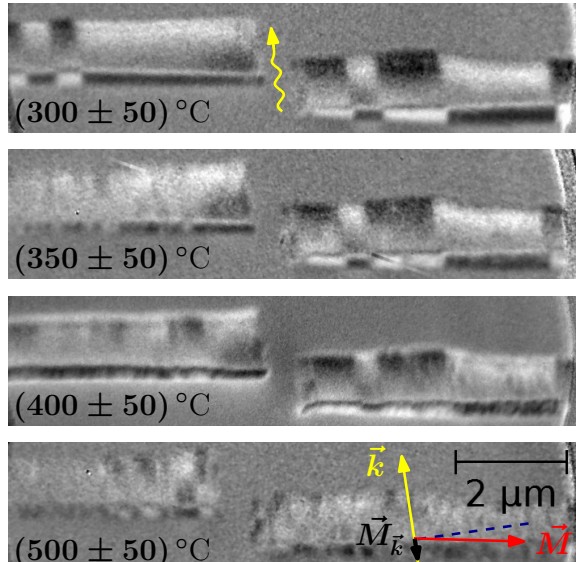

Fig. 4: **Changing the magnetic anisotropy upon annealing of the tubes**. XMCD-PEEM images (same contrast range [-13%..13%]) of the same tubes after annealing at increasing temperature. All images are taken after cooling down to room temperature. The X-ray beam arrives close to perpendicular to the tube axis as indicated by the arrow. With the increasing annealing temperature, the azimuthal magnetization pattern becomes weaker and gradually disappears, persisting only close to the tube extremity (end curling). The degree of the transformation is not the same for both tubes, probably due to a different tube wall thickness. After remagnetization along the tube axis (bottom image), both tubes display close to uniform contrast in the shadow, a sign of magnetization pointing along the tube axis. The contrast (bottom image) is weak due to a very small component of the magnetization along the beam.

tube axis). Other magnetization states compatibles with this magnetic pattern, are uniform transverse magnetization close-to-perpendicular to the beam direction and/or decrease of the magnetic moment. Both cases are highly improbable, as an external magnetic field would be required to sustain the uniform transverse magnetization in the whole tube. As regards magnetization, similar electroless-deposited materials are known to increase their magnetic moment upon annealing [32]. Azimuthal magnetization persists only at the ends of some tubes (e.g. Fig. 4, left tube, for 350°C annealing). Such so-called end curling state is expected from the locally high demagnetizing field [16, 17]. It has been recently observed by Wyss et al. at the ends of axially magnetized nanotubes [18].

Note that the degree of the transformation is not the same for all tubes for a given temperature (Fig. 4), possibly due to a slightly different tube wall thickness. Moreover, above 450°C annealing, some tubes exhibit defects – mainly holes (Appendix Fig. 11). These imperfections translate also into inhomogeneities in the magnetic configuration. During the annealing a few parameters affecting the anisotropy change (some of them are linked): the grain size increases (grain boundaries change) and the strain is reduced. Both effects are consistent with the reduction of the azimuthal anisotropy and thus presence of axial magnetization. Note that the magnetoelastic coupling itself can be affected by the annealing as well as the composition and crystallography.

# 6 Conclusion

We synthesized nanocrystalline CoNiB nanotubes by electroless plating in porous templates. Magnetic imaging revealed series of well-defined azimuthal domains, whereas previous theory and experiments all report axially-magnetized tubes. In our case the azimuthal anisotropy is promoted by an effective anisotropy coefficient of the order of $10\,kJ/m^3$, likely to originate from magnetoelastic coupling and/or anisotropic interfacial magnetic anisotropy. The strength of anisotropy and thus the magnetic configuration (axial, azimuthal domains) can be tailored through annealing or material composition. The CoNiB material is promising to search for predicted curvature-induced magnetic phenomena such as spin-wave-limited domain-wall motion, or the non-reciprocal propagation of spin waves.

# Acknowledgements

**Author contributions** S.S. and W.E. fabricated the samples and M.S. prepared them for measurements. L.C. did the transmission electron microscopy imaging. E.G. performed scanning transmission electron microscopy and chemical analysis (EELS). J.C.T. developed a code for XMCD-PEEM modelling (support for Fig. 2b). M.S., S.S., A.W., M.R., R.B., A.S., T.O.M., A.L., B.T., S.B., and O.F performed synchrotron XMCD-PEEM imaging. M.S., A.W., M.R., R.B., S.Y.M., and O.F. conducted the synchrotron STXM imaging. M.S. performed focused MOKE, magnetometry, electron microscopy+chemical analysis (EDX). M.S., A.W., and O.F. analysed and interpreted the data. M.S. and O.F. wrote the paper with contributions from S.S. and A.W. O.F. designed the project and supervised the work.

**Other contributions** The authors thank Márlio Bonfim and Jan Vogel for help with focused MOKE setup, Prof. Christina Trautmann (GSI Helmholtzzentrum for Heavy Ion Research) for the support with irradiation experiments for the polycarbonate template synthesis. The authors acknowledge Elettra and Soleil synchrotron facilities for allocating beamtime for the X-PEEM and STXM experiments and namely Sufal Swaraj, Stefan Stanescu and Adrien Besson

for STXM preparations (magnet, software, setup, . . . ).

**Funding information**    M.S. acknowledges grant from the Laboratoire d'excellence LANEF in Grenoble (ANR-10-LABX-51-01). S.S. gratefully acknowledge funding by the LOEWE project RESPONSE of the Hessen State Ministry of Higher Education, Research and the Arts (HMWK).

# A    Sample preparation

CoNiB tubes were prepared according to ref. [24], as shortly described below, by electroless deposition inside porous ion track-etched polycarbonate membranes.

## A.1    Electroless plating

Electroless deposition is a very flexible and powerful tool for the conformal coating of metal thin films on arbitrary substrates (even electrically non-conductive) [33, 34]. The deposition process is based on the autocatalytic reaction of the metal ions inside the plating solution at a specially functionalized surface. For the preparation of nanostructures, such as nanotubes, a template providing the proper shape is needed, such as an ion track-etched polymer foil. Both functionalization of the template and the plating itself are based on rather simple beaker chemistry.

## A.2    Chemicals

All glassware was cleaned with nitric acid and aqua regia before use. The solutions were prepared freshly with Milli-Q water ($> 18 \, \mathrm{M\Omega \cdot cm}$ at room temperature). The following chemicals were used without further purification: cobalt(II) sulphate heptahydrate (Sigma, 99.0%), dichloromethane (Promchem, 99.8%), borane dimethylamine complex – DMAB (Aldrich, pur 97%), ethanol (Labor Service GmbH, p.a.), potassium chloride (Merck, pur 99,5%), nickel(II) sulphate heptahydrate (Sigma, 99,0%), methanol (AppliChem, pure Ph. Eur.), palladium(II) chloride (Sigma, 99.9%), sodium citrate dihydrate (Sigma, puriss.), sodium hydroxide 32% in water (Sigma Aldrich, purum), tin(II) chloride (Merck, for synthesis), trifluoroacetic acid (Riedel-de Haën, $> 99\%$), and iron(II) sulphate heptahydrate (Sigma, 99%).

DMAB has rather low flash point of ($43 \, ^\circ\mathrm{C}$ / $109.4 \, ^\circ\mathrm{F}$), therefore it should be stored in cool place, preferably refrigerator.

## A.3    Template preparation

As a template, we used lab-made ion-track etched polycarbonate membranes with an average pore diameter around $300 \, \mathrm{nm}$ and length 30 microns (note that commercial membranes are also readily available). The track formation and track etching process is explained in literature [35]. For the synthesis of CoNiB and NiFeB nanotubes a $30 \, \mu\mathrm{m}$-thick polycarbonate (PC) foil (Pokalon from LOFO, High Tech Film GmbH) was irradiated with $Au^{26+}$ ions (fluence: $10^7$ ions/cm$^2$; kinetic energy of the projectile: $11.4 \, \mathrm{MeV}$ per nucleon) at the GSI Helmholtzzentrum für Schwerionenforschung GmbH (Darmstadt). The latent ion tracks were etched out at $50 \, ^\circ\mathrm{C}$ in 6M stirred sodium hydroxide solution for 11 min. The as-prepared template with cylindrical pores was washed with water and dried.

## A.4 Radial deposition in pores

First, the surface of the porous template is functionalized with Pd seeds to enable the metal deposition: the template is immersed for 45 min in a sensitizing $SnCl_2$-solution [42 mM $SnCl_2$ and 71 mM trifluoroacetic acid in methanol and water (1:1)], rinsed with ethanol and transferred for 4 min to activation $PdCl_2$-solution [11.3 mM $PdCl_2$, 33.9 mM KCl]. The process (sensitization+activation) is repeated for another two times (sensitization step reduced to 15 min instead of 45 min) to get more homogeneous layer of Pd catalysts. Afterwards, the template is washed with ethanol and water, and immersed in the plating solution for 20 min. The plating bath consists of 100 mM $NiSO_4 \cdot 7H_2O$, 30 mM $CoSO_4 \cdot 7H_2O$, 100 mM sodium citrate dihydrate, and 100 mM dimethylamine borane (DMAB). The deposition, reduction of metal ions by DMAB, takes place at room temperature and starts at the pore walls on the catalytic Pd seed particles and continues radially towards the pore centre (Fig. 5). Therefore, the shell thickness is controlled by the plating time. Note that the top/bottom surface of the template is covered as well. During the synthesis, hydrogen gas evolves at the template surface as a part of the deposition reaction.

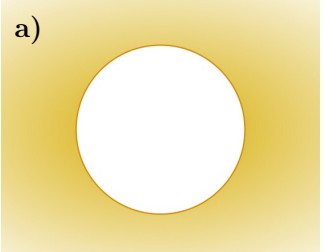
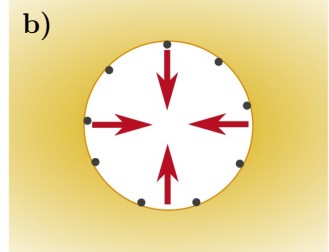
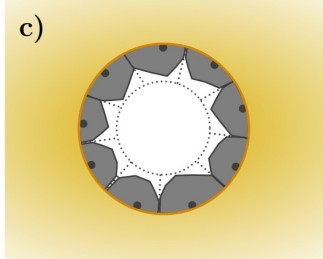

Fig. 5: **Scheme of the radial metal growth.** Cylindrical pore in the polycarbonate foil. a) Empty pore. b) Functionalized polymer surface through Pd-seeds on the pore walls. The arrows show the growth direction of the desired metal (Co, Ni). c) Metal grows radially from the pore interface inwards. The final states is indicated by the dotted lines.

## A.5 Preparation for measurements

After the deposition, washing and drying of the template, the metallic film deposited on the top and bottom surfaces of the polymeric template is removed by a gentle mechanic polishing (direction changed during the polishing) using a fine sand paper (e.g. SiC 1200/P-4000). As the measurement requires single (isolated) tubes, the polycarbonate template is dissolved in dichloromethane and the tubes are rinsed several times with the same solvent. Depending on the measurement technique, a droplet of solvent with the tubes is placed either on a doped Si substrate with Au alignment marks (for XMCD-PEEM), or on a Cu grid with a thin lacey carbon film (for transmission electron microscopies) or on a 100 nm-thick SiN membrane (for scanning transmission X-ray microscopy).

## A.6 Alignment of tubes on the substrate

As mentioned above, during the transfer of tubes from a solution (dichloromethane) onto the Si substrate, we used a magnetic field to align the tubes along given directions. The in-plane field is generated by a permanent magnet placed below the Si substrate. The orientation of tubes is further influenced by airflow in a chemical hood (note: dangerous solvent - dichloromethane), which is in our case predominantly parallel to the field direction. After evaporation of the solvent we can rotate the substrate and put another droplet with the solution to create a

second set of tubes aligned in a different direction. We used samples with orthogonal tubes or tubes aligned in one direction (Figure 6). These are crucial for imaging at Nanospectroscopy beamline in Elettra (Fig. 2a, 4), where the sample cannot be rotated on the microscope stage. In such case the orientation of tubes versus beam direction is set before mounting the sample by fine positioning of the substrate using an optical microscope. XMCD-PEEM imaging was also performed at Soleil Hermes beamline with a rotatable stage.

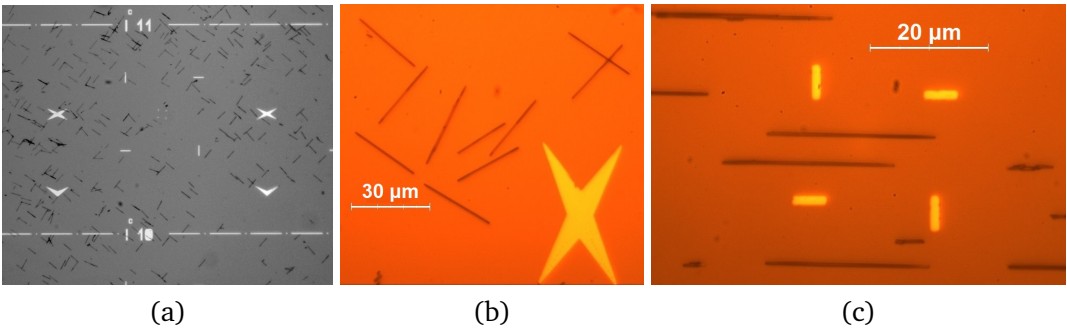

|  (a)  |  (b)  |  (c)  |

Fig. 6: **Optical images of magnetic CoNiB tubes dispersed on Si substrates with Au alignment marks**. (a) overview of a substrate with two orthogonal sets of tubes, (b) detail of another area (rotated by 90°). (c) Another sample, now with all tubes aligned along the same direction. Alignment of the tubes is promoted by a permanent magnet placed below the substrate during the dispersion of tubes from a solution.

# B  X-ray magnetic microscopies

## B.1  X-ray PhotoEmission Electron Microscopy (X-PEEM)

X-PEEM [36,37] experiments were performed at Nanospectroscopy (synchrotron Elettra) and HERMES (synchrotron SOLEIL) beamlines. The sample is irradiated with a monochromatic X-ray beam arriving 16° (18° at SOLEIL) from the substrate plane, with illumination size of several tens of microns. Excited photoelectrons are collected by PEEM from both sample surface and shadow area on the substrate (the latter arising from X-rays partially transmitted through the sample) [7]. Thanks to the grazing incidence of the beam, the resolution in the shadow is increased roughly by a factor of 3.6 ($1/\sin 16°$) along the beam direction. The energy of photons is tuned to the $L_3$ absorption edge of cobalt (around 778 eV). Circular magnetic dichroism, a difference in absorption of circularly left and circularly right polarized X-rays, leads to a difference in the photoelectron yield. The resulting contrast is related to the projection of magnetization along the beam direction. In the shadow area, which reflects volumic information integrated along the photon path, the situation is more complex and may require modelling [28]. The spatial resolution of X-PEEM is around 30-40 nm. The magnetic field was applied in-situ using a dedicated sample cartridge with a coil below the sample. Due to the collection of electrons, the technique is implemented under ultra-high vacuum.

## B.2  Scanning Transmission X-ray Microscopy (STXM)

This technique relies on the transmission of circularly polarized X-rays through a thin sample that must be placed on a thin, X-ray transparent substrate (100 nm-thick SiN membrane in our case). The X-ray beam is focused by diffractive Fresnel zone plate optics to a spot of 30 nm. Scanning by the sample (on piezo-stage) is performed in order to construct an image, pixel by

pixel. Magnetic imaging relies again on XMCD at Co-L$_3$. The contrast is very similar to the one obtained with XMCD-PEEM in the shadow. However, as this technique involves only photons, imaging under significant magnetic field is possible. We used STXM at HERMES beamline (synchrotron SOLEIL) to obtain images of CoNiB under variable axial magnetic field to extract the strength of the anisotropy field related to the azimuthal anisotropy. The magnetic field is applied thanks to a set of 4 permanent magnets whose orientation is controlled by motors. The setup enables application of magnetic field up to 200 mT. The imaging was conducted under primary vacuum (imaging under secondary vacuum or even atmospheric pressure is possible).

More information on both X-PEEM and STXM can be found in a review by Fisher and Ohldag [38] or book by Stöhr and Siegmann [39].

# C  Characterization of the tubes

## C.1  Chemical analysis

We used two techniques for chemical analysis of our tubes, namely Energy Dispersive X-ray Spectroscopy and Electron Energy Loss Spectroscopy. The former was employed in scanning electron microscope to probe sample area in tens or hundreds of nanometres. The later was used in scanning transmission electron microscope for analysis on the scale of few nanometres.

Chemical analysis by Energy Dispersive X-ray Spectroscopy (EDX) was conducted using different primary electron beam energies on both clusters and single tubes on a Si substrate as well as single tubes on a lacey carbon grid for transmission electron microscopy (TEM). Primary beam energies of 15 and 20 keV were used for the precise determination of ratio of metals (Co$_{80}$Ni$_{20}$), whereas much lower energies ($\leq$ 5 keV, namely 3.0, 4.5, and 5.0 keV) and single tubes on the TEM grid were used in order to detect boron B-K$_\alpha$: 183 eV (Fig. 7). Boron comes from the reducing agent used during the deposition. It influences the microstructure of the deposit, with more boron leading to finer grains and eventually to amorphous material [40]. As boron is quite light element, specific conditions and instrument setup are needed for B detection in EDX [41].

The boron composition seems to be around 10%, but we could not obtain reliable and precise results with EDX due to very low counts on the detector. In the literature, X-ray Photoelectron Spectroscopy (XPS) on significantly larger tubes (same concentration of the reducing agent) suggested a negligible B content [24], while Richardson et al. [32] found with XPS around 25% at of boron in electroless-deposited tubes using the same reducing agent (DMAB). They measured similar content for different deposited metals and alloys, concentration of metallic salts in the bath. The Boron content increased with lower pH of the plating bath; it should be also influenced by the concentration of the boron containing reducing agent (DMAB). In our case, on one hand the concentration of boron species in solution was lower (decreases B in the deposit), on the other hand, the pH of the bath is slightly lower (increases B in the deposit). Altogether we expect a (slightly) lower amount of boron in our tubes than reported by Richardson (25% at).

Aside from above-mentioned elements (Co, Ni, B), sometimes traces of Pd (seed particles in the deposition) and Sn (template modified with Sn(II) species) were detected with EDX as well. The presence of C and O is attributed mainly to the dissolution of the polymeric template, TEM grid with the C film, and unavoidable partial carbon contamination and surface oxidation.

Electron Energy Loss Spectroscopy (EELS) reveals grains (clusters) with grain boundaries being rich in light elements, including oxygen (for samples exposed to the air).

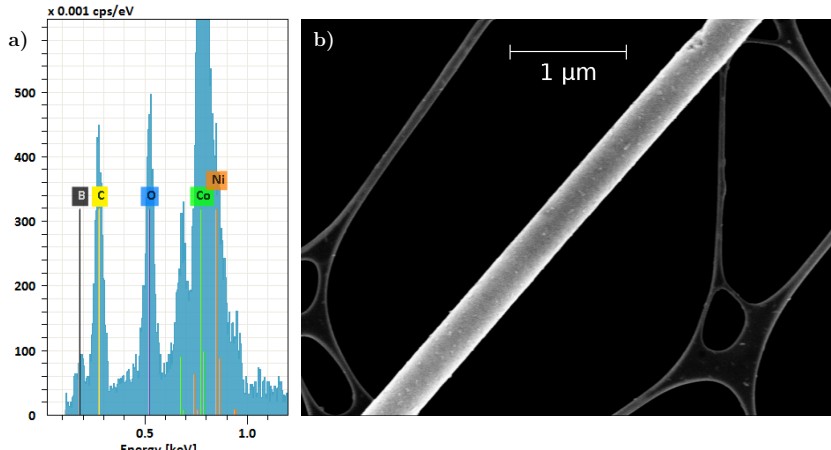

Fig. 7: **Chemical analysis using EDX.** a) EDX spectra acquired with low primary beam energy (4.5 keV) showing the boron presence aside from expected Co and Ni, as well as C and O coming mainly from the template dissolution and possibly partial tube oxidation for the later element. b) Electron microscopy image of the investigated tube on a lacey carbon film. EDX spectrum taken in the middle of the tube. Similar results were obtained at different points as well as when averaging over larger tube area.

## C.2 Magnetometry on tube array

Fig. 8a shows a hysteresis loop obtained by VSM-SQUID for an array of CoNiB tubes in a polycarbonate matrix with magnetic field applied along the tubes. The pore density is very low (Fig. 8b) and the hollow nature of tubes reduces the total magnetic moment compared with wires of identical diameter, so that we expect weak magnetostatic interactions, contrary to the case of anodized alumina templates (significantly higher pore density) and solid nanowires [42].

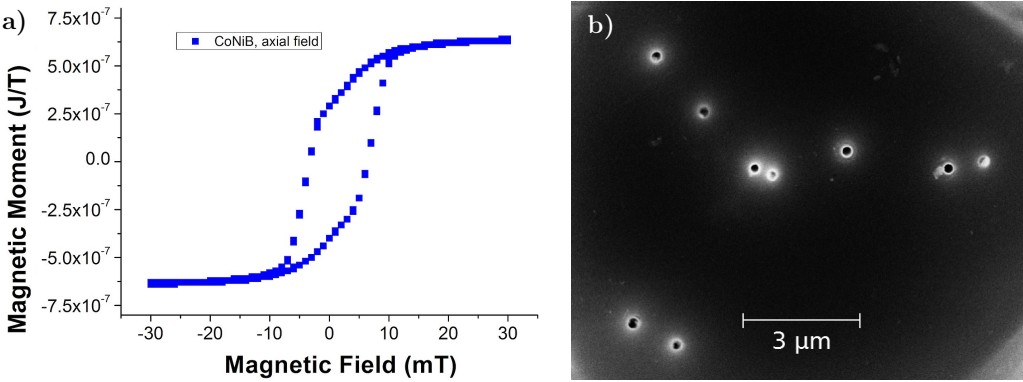

Fig. 8: **Magnetometry on array of tubes.** a) Hysteresis loop on a sparse array (low density, low interactions) of CoNiB tubes in the polycarbonate template, measured by VSM-SQUID. The field is applied parallel to tube axis. b) Scanning electron microscopy, top view of a part of the measured template illustrating the low density of pores.

Note that the hysteresis loop obtained on the array of tubes (still in the template) is rather square, with significant remanence. This contrasts with the measurement on tubes isolated from the template, where X-ray microscopies displayed magnetic states (azimuthal domains)

with very low remanence. However, other loops acquired with focused magneto-optics on isolated tubes were slanted, unlike the ensemble of tubes. Therefore, aside from the possible interactions between the tubes we cannot rule out that liberation of tubes from the template and laying them on a supporting substrate can alter their properties. Note that the template with array of tubes is already polished, thus the difference is not caused by polishing-induced strain. On the other hand, the polishing is not perfect and parts of the top/bottom film on the template are still present (few patches visible also in Fig. 8b). Despite this unpolished area being small, we cannot rule out its contribution.

### C.3  Magneto-optics with focused laser

Aside from the tube arrays, we also measured isolated tubes dispersed on a Si substrate via the Magneto-Optical Kerr Effect (MOKE), implemented in the longitudinal configuration with a focused He-Ne ($\lambda = 632.8$ nm) laser (spot 1 $\mu$m). The field was swept as a triangular wave signal, with frequency 1.1 Hz, and field calibration uncertainty max $\pm 5$ mT. The maximum laser power was 1.1 mW, for some measurements we used just 0.2 mW. Fig. 9 shows hysteresis loops obtained on the tubes. Some loops are slanted (Fig. 9a) with almost no remanence (tube axis = hard axis for magnetization), which is consistent with the synchrotron data, where magnetization is azimuthal at remanence (perpendicular to the tube axis) and under axial field gradually rotates towards the axis (see Fig. 3 in the main text). However, in some cases (different tubes, even different tube part in one instance) the loops are quite squared (Fig. 9b).

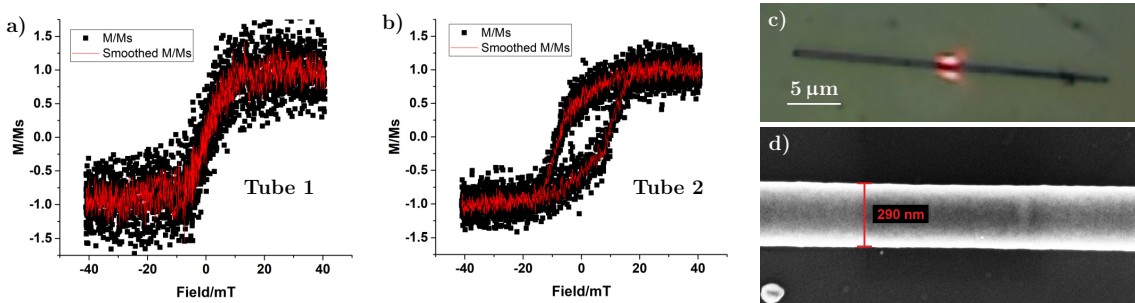

Fig. 9: **Magnetometry on a single tube – magnetooptics with focused laser.** Hysteresis loops for the axial magnetic field: a) slanted curve, b) rather squared loop obtained on another tube. Average of 100 loops with short acquisition time (0.9 s). Data processed in OriginPro: line subtracted, normalized, smoothed: average of 7 adjacent points (red curve) - original curve 5000 points (black points). c) Optical image (magnification 100x) of a tube with the diffracted laser spot, the magnetic field is applied in the horizontal direction, close to parallel to the tube axis. d) Scanning electron microscopy image of a central part of the tube from c).

This was also the case when one tube (previously non-irradiated) was probed with just 0.2 mW laser power – close to the minimum power we can apply and measure some magnetic signal from tubes in our setup. We assume that the loop squareness is caused by laser heating due to a bad thermal contact with the substrate for some tubes. As tubes are dispersed from a solution, template dissolution products may create a halo around structures and decrease the thermal conductivity of the contact. Further, in case of tubes (cylindrical objects in general) the contact area is rather small. Hysteresis loops acquired with higher laser power show larger squareness. Only several tubes were measured with MOKE compared to several tens of tubes investigated by higher resolution X-ray microscopies. Therefore, in the determination of the anisotropy strength we rely on the X-ray microscopy (STXM under field).

# D   Possible microscopic sources of azimuthal anisotropy

As our tubes are cobalt-rich, the first contribution coming to mind is the magnetocrystalline anisotropy. However, as our tubes are nanocrystalline with a random texture (see Fig.1), we rule out the magnetocrystalline contribution. There must be another source of magnetic energy, for which the degeneracy between the axial $\hat{z}$ and the azimuthal $\hat{\phi}$ direction is lifted. While both directions are normal to the radial direction $\hat{r}$, and are thus locally similar to the two in-plane directions for a thin film, the difference is the existence of curvature along the azimuthal direction. We consider below two possible sources of magnetic anisotropy that could arise from the direction-dependent curvature: intergranular interface anisotropy and magnetoelasticity. As mentioned in the main text, owing to the radial growth process, grains are expected to have their shape and size varying differently along the two directions. We detail below handwaving models, and show that both sources may provide a strength of anisotropy whose order of magnitude is consistent with the experimental data.

First we discuss the interface anisotropy. As our samples are nanocrystalline, the proportion of atoms in the vicinity of a grain/cluster boundary is not negligible, so that interface anisotropy $K_s$ with e.g. boron-rich grain boundaries could arise. We consider a tube with outer diameter 250 nm and wall thickness 25 nm. Assuming an isotropic grain size $l_0$ upon grain nucleation from the outer diameter, the azimuthal grain size $l_\phi$ at the inner diameter should be reduced by 20 % (outer radius 125 nm, inner one 100 nm and thus (125 nm-100 nm)/125 nm=0.2; the grain size along the azimuth is directly proportional to the radius). We further assume that the grain size along the tube axis $l_z$ stays constant. Thus on the average along the radius the anisotropy of grain size $\delta = \frac{\langle l_z \rangle - \langle l_\phi \rangle}{(\langle l_z \rangle + \langle l_\phi \rangle)/2}$ is 0.1, yielding a slightly wedge-shaped grain (such as in Fig. 5c); $\langle l_\phi \rangle = 0.9 l_0$, $\langle l_z \rangle = 1 l_0$ being average grain sizes along the azimuth and the tube axis, respectively, and $l_0$ is the grain size on the outer surface. Transmission electron microscopy suggests that the grain size is of the order of $t = 10$ nm. The anisotropic contribution $K_{\text{eff}}$ of $K_s$ to the effective volume magnetic anisotropy is therefore $2\delta K_s / t$. Considering $K_s \approx 0.2$ mJ$\cdot$m$^{-2}$ as an estimate (values nearly one order of magnitude higher may exist at some interfaces, for instance between 3d elements and some oxides [43]), one finds: $K_{\text{eff}} \approx 4 \times 10^3$ J$\cdot$m$^{-3}$. Expressed in anisotropy field: $H_{\text{eff}} \approx 2K_{\text{eff}}/(\mu_0 M_s) = 8 \times 10^3$ A/m, or: $\mu_0 H_{\text{eff}} \approx 10$ mT. This is of the same order of magnitude as the measured value of 25 mT. Another contribution of interface anisotropy may be due to the curvature of the outer and inner parts of the grains, so that the orientation of atomic bounds is on the average slightly different along the axial and azimuthal directions. A modelling would however require advanced information about the structure of the interface, which is not available.

We now discuss the magnetoelastic anisotropy. Borides of 3d ferromagnetic elements are known to display sizeable magnetoelastic effects [31] (except for $(Ni_{80}Fe_{20})B$ and CoFeNiB with certain compositions with almost zero magnetostriction; NiFeB tubes are described below), and electroless plating is also known to deliver strained materials. It is probable that the expected wedged shape of the grains described above (see also Fig. 5c) induces a building of a higher compressive strain $\epsilon$ along $\hat{\phi}$ while the grain grows inward, because there is less and less space to accommodate incoming atoms. The saturation magnetostriction of Co-rich CoNi borides is of the order of $\lambda \approx -6 \cdot 10^{-6}$, more values with references can be found in Tab. 1. For 3d metals the combination of elastic coefficients $c_{11}-c_{12}$ is of the order of $10^{-11}$ N$\cdot$m$^{-2}$, or $10^{-11}$ J$\cdot$m$^{-3}$. Thus, the linear magnetoelastic coefficient is $B_1 \approx -10^6$ J$\cdot$m$^{-3}$. An anisotropy of strain of 0.4 % would therefore be required to account for the observed microscopic anisotropy.

To conclude this part, electroless-grown materials are expected to develop nanograins with some anisotropic structure features along the azimuthal and axial directions, associated with

the local curvature of the supporting surface, be it shape or strain. A resulting contribution to magnetic anisotropy is expected, which could arise from both interface anisotropy and magnetoelastic coupling. Realistic figures show that both sources are consistent to explain experimental results. Without further knowledge on the structural anisotropy of the nanograins, which would be challenging to access, it is not possible to decide unambiguously which phenomenon is dominating.

Table 1: **Saturation magnetostriction $\lambda_s$ for some Co-rich CoNiB compounds.**

| material | $\lambda_s$ | reference |
|:---:|:---:|:---:|
| $(Co_{80}Ni_{20})_{80}B_{20}$ | $-5 \cdot 10^{-6}$ | [31] |
| $Co_{80-x}Ni_xB_{20}$ | $-7 \cdot 10^{-6}$ for $x \in (0; 12)$ | [44] |
| $(Co_{80}Ni_{20})_{77}B_{23}$ | $-8 \cdot 10^{-6}$ | [45] |

# E NiFeB tubes with axial magnetization

As mentioned in the main text, $(Ni_{80}Fe_{20})B$ tubes (diameter 350-390 nm) were fabricated using the same electroless deposition technique and templates, only cobalt (II) sulfate in the plating solution was replaced by iron (II) sulfate. We found out that tubes of this material are axially magnetized (Fig. 10). These tubes were grown also in confined pores and the growth proceeds radially, therefore similar strain and grain structure could be expected. However, unlike CoNiB, these NiFeB tubes have almost zero magnetostriction [31] and therefore the magnetoelasticity is negligible. In addition, Fe-based alloys are also known to display lower interfacial anisotropy. In other words, both above-discussed anisotropy sources are expected to be weaker in magnitude, being consistent with axial magnetization as expected from a soft magnetic material.

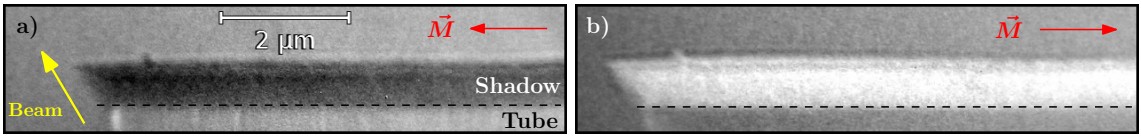

Fig. 10: **XMCD-PEEM images (Fe-L$_3$ edge) of an electroless-deposited NiFeB shell** magnetized axially sequentially along two opposite directions. The beam arrives from the bottom in a direction depicted by the arrow. Only the shadow area (information from the volume) is clearly visible due to selected focus on the shadow and partial oxidation of the outer tube surface. The tube is axially magnetized with magnetization component a) parallel and b) anti-parallel to the X-ray beam. The magnetization was switched by applying 16 mT close to the axial direction (tilted with the beam). Switching field of these tubes is 10-16 mT.

# F Annealing of CoNiB tubes

The in-situ annealing of CoNiB tubes was performed under ultra-high vacuum, however, in a chamber distinct from the X-PEEM microscope chamber. We ramped the temperature to the desired value, keeping it at least for 30 min (except for the first one, 300°C – only 10 min),

and then we let the sample cool down to room temperature. The annealing was repeated several times with a gradual increase in the target annealing temperature. The imaging was performed after each annealing step. The temperature control was not very precise as we used a small current-heated filament below the sample. The temperature was estimated based on previous and similar filament heating experiments, and on a comparison with annealing of twin samples in a more controlled environment (high vacuum furnace). This implies an uncertainty of $\pm 50\,^{\circ}$C.

## F.1  Defects upon annealing

Upon annealing under vacuum, hollow defects appeared in the shell of some CoNiB tubes, for temperatures typically above $450\,^{\circ}$C. These holes are visible both in X-PEEM and subsequent scanning electron microscopy images (Fig. 11a, here an extreme case is shown). Not all tubes had the same density of holes upon the annealing (Fig. 11b), which may come from variation in the tube wall thickness. Some tubes do not display any visible damage.

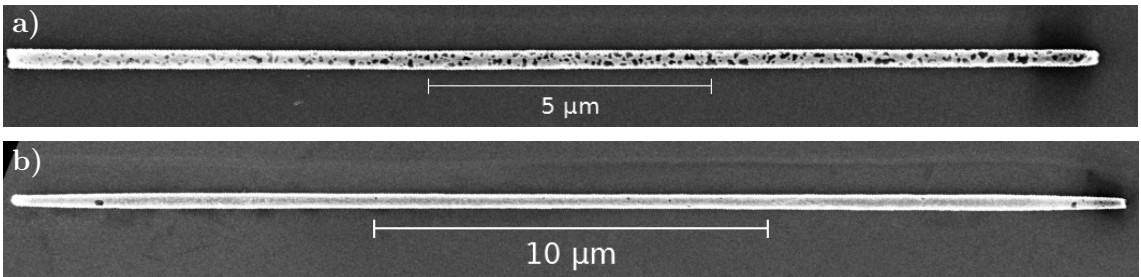

Fig. 11: **Defects upon in-situ annealing** ($500\,^{\circ}$C$\pm 50\,^{\circ}$C). SEM images of two tubes lying on the same substrate, displaying a very different amount of defects after annealing. Both tubes displayed axial magnetization after annealing. The difference may come from a variation in the tube wall thickness.

As the calibration of temperature for the in-situ annealing is not accurate, for comparison we performed annealing experiments in a separate vacuum furnace with a better control of temperature as both the substrate and the environment are at the same temperature. Now we will briefly mention possible differences in experimental conditions between annealing done inside the preparatory chamber of the PEEM setup (in-situ annealing) and the vacuum furnace annealing. However, we do not suppose that they play a significant role. The PEEM preparatory chamber is operated under ultra-high vacuum. However, during the annealing the pressure increases substantially and it is of the same order of magnitude as the pressure in the vacuum furnace (secondary vacuum, $< 10^{-4}$ Pa). The main difference might be X-ray beam irradiation of some tubes before the annealing, in particular effect of the X-rays on the tubes and impurities that cover them (breaking bonds, graphitizing hydrocarbons, etc.). As only part of the sample was irradiated, but the whole sample was annealed, we could conclude that there is no big difference between irradiated tubes and tubes not exposed to X-rays (based on electron microscopy images of both sets of tubes). As we used twin samples on the same substrates in both (in-situ, furnace) annealing experiments, we suppose that both are comparable.

Even the furnace annealing (at least 30 min, secondary vacuum) provided tubes both with and without significant defects for temperatures $450\,^{\circ}$C, $500\,^{\circ}$C, $550\,^{\circ}$C, and $600\,^{\circ}$C. Still more defects in larger amount of tubes appear with increasing temperature, especially above $550\,^{\circ}$C. For a lower temperature, $400\,^{\circ}$C, no significant defects were present, but on the other hand the transformation to axial magnetization was not complete. At $550\,^{\circ}$C most of the tubes are severely damaged with many holes, only a minority of tubes is rather intact and some tubes

survive also up to 600 °C. Therefore, the maximum annealing temperature before significant defects appear seems to be 450 °C − 500 °C. Further, we tried shorter (15 min) and longer (150 min) annealing time for 450 °C. 15 min led to almost no defects, but the increase of the grains size with respect to the as-deposited sample was very small, suggesting that longer annealing is needed. Longer (150 min) experiment produced slightly more defects such as tubes broken in places where there were already some small defects. The presence of larger defects (especially above 400 °C) can be an issue as they lead to inhomogeneity in the magnetic configuration. We tried to tackle this problem fortifying the tubes with an additional inner (non-magnetic) layer deposited either by electroless plating or atomic layer deposition (ALD). It seems that the amount of defects in such tubes upon annealing is lower. Alternatively it is possible to perform the ALD after dispersion of tubes on the substrate – this improves not only mechanical stability, but also protects the tubes from further oxidation. But we refrained from such treatment as the electrically-insulating oxide cover layer can cause problems (charging) in collecting photoelectrons in XMCD-PEEM.

### F.2   XMCD-PEEM: reversal of in-situ annealed tubes

After the in-situ annealing, magnetization of the CoNiB tubes is longitudinal. We used a coil fitted in the XMCD-PEEM sample cartridge to apply magnetic field to these tubes. A few mT applied along the tube axis were sufficient to fully reverse the magnetization direction (Fig. 12). The contrast on the tube as well as in the shadow is weak, due to the close-to-perpendicular beam orientation with respect to the tube axes and thus the magnetization direction.

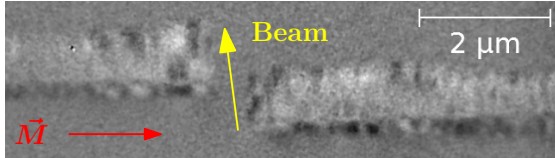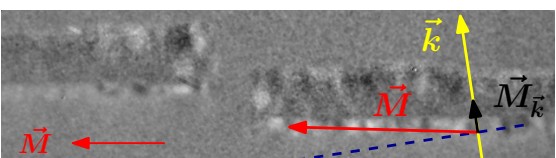

Fig. 12: **Magnetic switching of annealed tubes with axial magnetization**. The magnetization in the axially-magnetized tubes can be reversed by applying field along the tube axis, as seen from the left to the right image. The beam arrives from the bottom of the image, and is close to perpendicular to the tube axes, so that the projection of magnetization to the beam direction is small. This leads to weak magnetic contrast. Still one can distinguish the switch, both on the tube and in the shadow. In both images the left tube displays some azimuthal curling close to its end, as seen in the shadow. Both tubes display several defects (holes) due to over-annealing.

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
