# Peer review of "Flux-closure domains in high aspect ratio electroless-deposited CoNiB nanotubes"

_SciPost Physics, doi:SciPost Phys. 5, 038 (2018)_

## Round 2 · Referee Report · Anonymous (Referee 1) · 2018-8-3

Strengths

  1. complete study of magnetic nano tubes from the fabrication to magnetic imaging.
  2. combination of two x-ray microscopy techniques and comparison to MOKE.

Weaknesses

  1. A comparison between simulations and experiments would give a more detailed insight into the domain and domain wall structure.
  2. The domain wall structure is not shown.
  3. access to 3D information of the domains would be interesting.

Report

The Authors describe the fabrication, characterization and magnetic study of ferromagnetic nanotubes. They introduce the material systems and the imaging techniques used. They present the results clearly and draw the correct conclusions.

Requested changes

-

  • validity: high
  • significance: good
  • originality: high
  • clarity: high
  • formatting: excellent
  • grammar: excellent

Author:  Olivier Fruchart  on 2018-08-23  [id 309]

(in reply to Report 1 on 2018-08-03)
Category:
answer to question

We appreciate the comment of the referee.

In the present manuscript we wished to report on a new material, displaying spontaneously azimuthal magnetization. Comment 3, and partly comment 1, are directly relevant for this concern. We fill that this comment is addressed in the manuscript by the following statement (page 7): "it is difficult to extract quantitatively the direction of magnetization in this series, because of the exponential decay of photon intensity inside matter, uncertainties in the dichroic coefficient, and the existence of a background intensity in the image. We can only provide an estimate of the $H_\mathrm{K}$ from the field for which all contrast vanishes in the corresponding images". While we could always attempt to reproduce the experimental STXM contrast, we fear that the several sources of uncertainties would prevent us from gaining more information than the fact that magnetization is azimuthal.

Comments 1 and 2 pertain to the domain wall structure. This is a very interesting focus, crucial for domain-wall motion under field or current. This topic is quite rich and not so simple. We have significant experiments and simulations on this topic, which we will put together in a dedicated manuscript. The aspects of the two comments will be largely covered.

---

## Editorial Decision

published